# HESS Opinions: The Sword of Damocles of the Impossible Flood

Alberto Montanari[1], Bruno Merz[2], Günter Blöschl[3]

[1] Department of Civil, Chemical, Environmental and Materials Engineering, University of Bologna, Bologna, Italy,
[2] Section Hydrology, GFZ German Research Centre for Geosciences, Potsdam
Institute for Environmental Sciences and Geography, University Potsdam
[3] Institute of Hydraulic Engineering and Water Resources Management, Technical University Vienna, Vienna, Austria

*Correspondence to*: Alberto Montanari (alberto.montanari@unibo.it)

**Abstract.** Extremely large floods that far exceed previously observed records are often considered virtually 'impossible', yet are an ever-present threat similar to the sword suspended over the head of Damocles in the classical Greek anecdote. Neglecting such floods may lead to emergency situations where society is unprepared, and to disastrous consequences. Four reasons why extremely large floods are often considered next to impossible are explored here, including physical (e.g. climate change), psychological, socio-economic and combined reasons. It is argued that the risk associated with an 'impossible' flood may often be larger than expected, and that a bottom-up approach should be adopted that starts from the people affected and explores possibilities of risk management, giving high priority to social in addition to economic risks. Suggestions are given for managing this risk of a flood considered impossible by addressing the diverse causes of the presumed impossibility.

## 1 A flood considered virtually impossible

In July 2021 heavy and widespread rainfall caused catastrophic flooding in parts of Germany, Belgium, Luxembourg and the Netherlands with 221 fatalities and € 46 billion economic damage (Munich Re, 2022). A total of 134 people died alone along the Ahr River in Germany, a tributary of the Rhine with a catchment area of less than 900 km². Although a precise rainfall forecast was only possible with a lead time of a few hours, the potential for extraordinary rainfall was provided at least two days in advance (Mohr et al., 2023). Nevertheless, this flood took both people and disaster management by surprise. The severity of the flooding was underestimated, the state of emergency was declared too late and evacuations were not initiated in time (Thieken et al., 2023), although the specific circumstances of the warning and disaster management are still under investigation by enquiry commissions (Landtag RLP, 2023). The first fatalities in the Ahr catchment occurred in the late afternoon of the 14th of July in Dorsel in the upstream Ahr catchment. The flood wave took around seven hours to travel downstream, and at 2 a.m. the next day 12 people in a facility for mentally and physically handicapped were trapped and drowned in Sinzig, 60 km downstream of Dorsel. In many regions, the inundation extent of the July 2021 exceeded the flood

scenarios of the official flood maps by far. For instance, in the city of Bad Neuenahr-Ahrweiler, where 67 people died, the 2021 flood extent was four times larger than the extreme scenario associated with a return period of 200-300 years, and

seven times larger than the 100-year scenario. Figure 1 shows the time series of the annual maximum flow of the Ahr River at Altenahr. Clearly, locals were little prepared for an event of this magnitude.

## 2 What is an impossible flood?

Our perception and preparedness for river floods is dictated by experience, and thus by the occurrence of past events. In flood risk management, communities often design their management strategies based on historical events. In fact, international and

national regulations recommend design recurrence intervals that usually do not exceed 100 or 200 years, while impossible floods are far less frequent. For example, the EU Flood Directive (Commission of the European Communities, 2007) requires member states to map inundation areas for an extreme scenario which has typically a return period of 200 years. Large events, or different types of events, are often considered essentially 'impossible' because they exceed the expectations based on historical experience. Such impossible events include floods whose probability of occurrence is considered too small to

act, but also events that go beyond any imagination. These events are termed here 'impossible floods', to highlight a common thread of thinking, even that of flood managers. Impossibility is felt despite the scientific awareness that no natural disaster, even an exceptional one, can in principle be considered impossible.

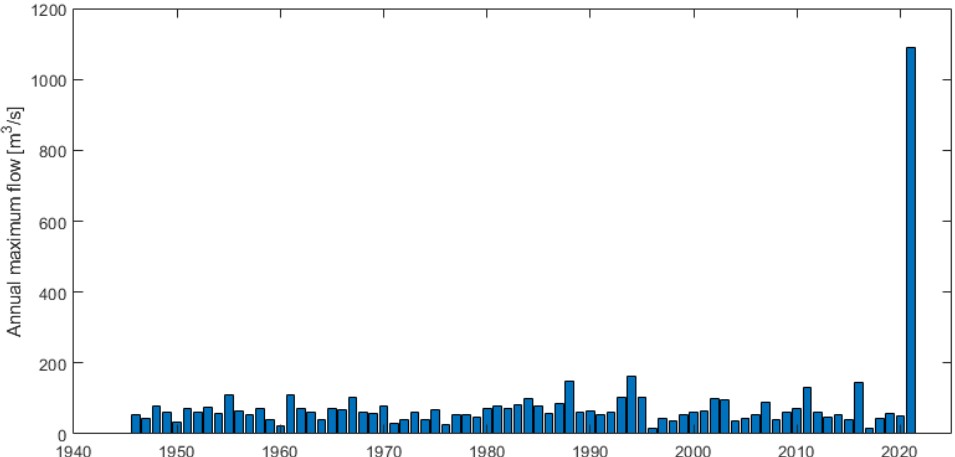

**Figure 1: Time series of the annual maximum flow of the Ahr River at Altenahr from 1946 to 2021. Data: Landesamt für Umwelt Rheinland-Pfalz**

From a technical point of view, the underestimation of the impossible flood is caused by the use of observations to calibrate models for estimating extreme events, which in turn instruct technical standards and risk mitigation and adaptation strategies. Use of historical information is often made with the (sometimes tacit) assumption that the future will repeat itself in a similar

way as the past, for the simple reason that this assumption is the most obvious choice in the absence of other information. However, in reality the future may be quite different from the past, not least because of climate change and our limited knowledge of the past itself. The future may thus have in store a flood considered impossible, a Sword of Damocles, an ever-present but not perceived threat (Sofia et al., 2017; Bloeschl et al., 2019; Bloeschl et al., 2022).

According to legend, Dionysius I, tyrant of Syracuse, suggested to his courtier Damocles that he take his place for a day to
make him realise that the happiness of a ruler is very precarious. In the evening, Damocles attended a banquet and, at the end of the meal, realised that a sword hung from the ceiling held only by a horsehair, a symbol of the danger hanging over the rulers (Fig. 2). The situation of many communities is similar: they are exposed to floods with very low probability but high damage potential. This combination of low probability and high damage is thus sometimes referred to as the Damocles risk type (GACGC, 2000), which is shown in the context of other risk types in Fig. 3. The risk associated with Damocles-type
events is often disregarded, planning for a prosperous socio-economic development while the Sword of Damocles of the impossible flood hangs by a thread whose already intrinsic weakness is accentuated by climate change and other causes that we will discuss later.

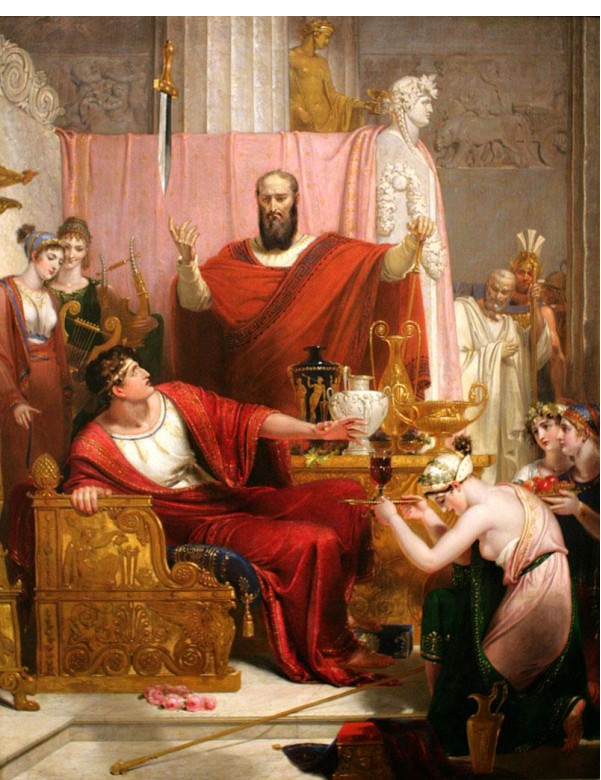

**Figure 2: Damocles notices the sword above him. Richard Westall, Public domain, via Wikimedia Commons.**

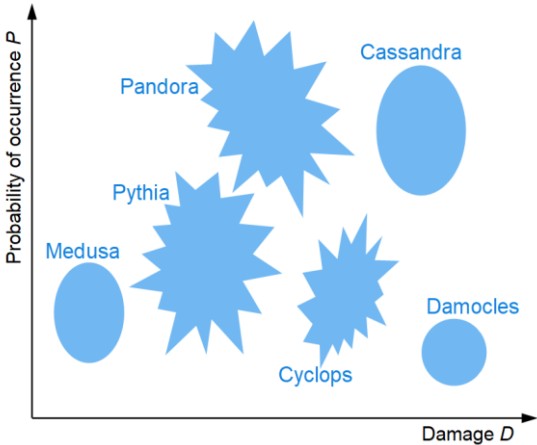

**Figure 3: Probability of occurrence (P) and damage (D) of different types of risk. Damocles: P very low, D very high, e.g. extreme floods, meteorite impact. Cyclops: P not well known, D high, e.g. earthquakes. Pythia: P not well known, D largely unknown, e.g. releasing transgenic plants. Pandora: P not well known, D not well known but persistent and even irreversible damage expected, e.g. persistent organic pollutants. Cassandra: P high, D high, e.g. destabilization of ecosystems. Medusa: P low, D low, e.g. electromagnetic fields. Starlike shape indicates uncertainty of estimating P and/or D. Modified from GACGC (2000).**

The 2021 flood in Germany was certainly a Damocles-type event, a flood that until then had been considered next to impossible to occur by the communities and citizens. Numerous other examples exist that were larger than those known to the locals, sometimes occurred faster, or had larger spatial extents, such as the November 1966 flood of Florence (Galloway et al., 2020), and the August 2002 flood of the Kamp in Austria (Merz and Blöschl, 2008a, 2008b). Damocles-type events may also involve compounding or cascading effects, such as the September 2022 flood in the Marche Region of Italy which mobilized large masses of woody debris that clogged bridges thus reducing river conveyance. With the benefit of hindsight, the above-mentioned events had a clear cause and do not appear impossible. Understanding these causes better may assist in getting better prepared (Merz et al., 2015, 2022).

## 3 Reasons why rare events are considered impossible

There are numerous reasons why extreme events such as the Ahr flood are considered impossible prior to their occurrence, resulting in a lack of preparedness. They can be classified into (a) physical, (b) psychological, (c) socio-economic and (d) combined reasons (Merz et al., 2015) as illustrated in Fig. 4.

### 3.1 Physical reasons

Physical reasons are inherent climatic and hydrological features of the processes causing the event that make it difficult to predict based on available information. They include non-linearity, randomness and changes of the process dynamics. Non-linear behaviour occurs, for example, if a small increase in precipitation causes saturation of a large part of the catchment which may give rise to an unexpectedly extreme river flow (Rogger et al., 2013; Vreugdenhil et al., 2022). This was the case

for the Kamp flood in 2002, when a moderate increase in rainfall led to a disproportionate and sudden increase in river flow, which was unexpected based on the information available at the time. Another example is given by the flood that occurred in the Romagna region, in Italy, in May 2023. Here, a first extreme storm induced high antecedent soil moisture for a subsequent storm that caused extreme runoff and then extensive landslides and flooding. Strong non-linearities are also

observed in the impacts, for instance, when local flooding triggers large-scale abrupt failures of infrastructure networks (Wang et al., 2019). Randomness is another factor contributing to the complexity of the system, in particular, when the probability distribution of the flood magnitudes at the upper end decays slower than exponential, a behaviour referred to as "heavy tailed distribution". In other words, the probability of extreme events is higher than usually expected. Merz et al. (2022) proposed three main reasons for heavy tails:

- Mixture of storm types, for instance, tropical cyclones (Villarini and Smith, 2010) that produce very high flood peaks with low probability, and frontal rain storms that produce smaller floods with larger probability;

- Non-linear rainfall-runoff response, in particular if hydrological thresholds such as saturation are exceeded;

- Random process interactions, such as unlikely but possible combinations of high rainfall and high runoff coefficients in dry climates (Viglione et al., 2009).


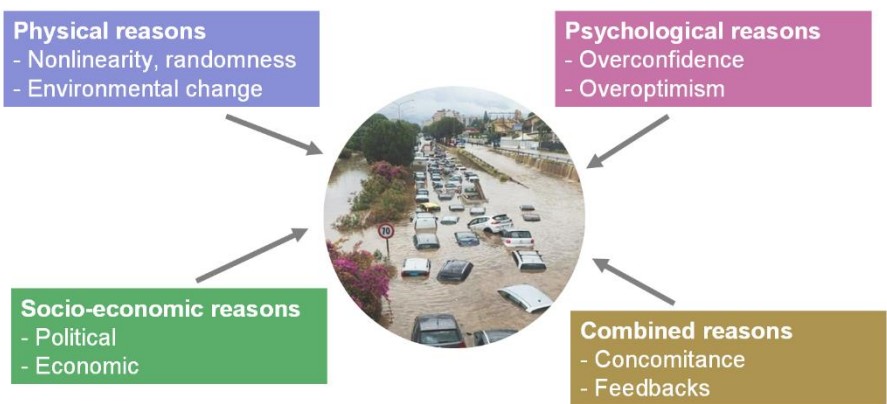

**Figure 4: Reasons why extreme floods may be deemed impossible: The complexity of physical features makes events unpredictable. Psychological factors mean that the risk is not objectively assessed. Socio-economic dynamics mean that the measures deemed necessary are not implemented. The combined reasons further increase the complexity.**

Regarding changes of process dynamics, Blöschl (2022) highlighted three main factors that affect the flood regime:

- Land use changes, including deforestation, urbanisation and soil compaction by agriculture, which are particularly relevant in small catchments, as soil permeability plays an important role in regulating infiltration at this scale.

- Hydraulic interventions, including river regulation, levees and dams, which have the greatest impact on medium-sized events, as more extreme events generate widespread flooding irrespective of these interventions.

• Climate change impact, which depends on the environmental context. In large catchments, the seasonal interaction between soil moisture, snow melt and regional precipitation are most relevant, while air temperature is not a direct predictor of change. In small basins, which have shorter concentration times, short-duration, high-intensity thunderstorms (often of convective origin) are more relevant and these tend to increase with air temperature, thus exacerbating the risk of flooding in a warmer climate.

**3.2 Psychological reasons**

Inconsistency between personal perception and reality, i.e. cognitive distortion, is another reason that may contribute to the belief that extreme events are impossible. Some of these distortions may cause underestimation of the flooding potential and consequences. Other distortions may lead to overconfidence in the understanding of the system. In the case of the Vajont disaster, for example, when a massive landslide into the reservoir caused a flood wave that overtopped the dam and led to

nearly 2,000 fatalities, overconfidence in estimating the landslide dynamics contributed to underestimating the risk of the disaster (Delle Rose, 2012). Experimental evidence shows that overconfidence is a widespread phenomenon (Hammitt and Shlyakhter, 1999). The cognitive distortions listed below, taken from Merz et al. (2015), may contribute to incorrect risk estimation and risk management:

• Erring on the side of least drama: Calling for higher levels of evidence to support dramatic and alarming
conclusions.

• Recency bias: Overemphasizing the importance of recent events, e.g. associating a higher probability of occurrence with more recent events, even when information of large past events is available.

• Illusion of certainty: Emotional need for certainty when it does not exist.

• Overconfidence: Overestimation of one's own knowledge.

• Retrospective bias: Overestimation of the probability with which known outcomes would have been correctly predicted.

• Single-action bias: Relying on one action, even when this action reduces the risk only marginally. Therefore, a significant risk (residual risk) remains that is often not perceived (CRED, 2009). Residual risk is defined as the risk that remains, even when effective disaster risk reduction measures are in place, and for which emergency response
and recovery capacities must be maintained (UNISDR, 2009; Fiori et al., 2023).

Often, several biases occur at the same time. For instance, people tend to consider the flood problem as solved as soon as structural protection has been built (single-action bias). This action reduces their concerns, thus ignoring the often substantial, residual risk. At the same time, known historical floods that would overwhelm the existing flood defences are perceived as less relevant as they have occurred in the distant past (recency bias). For instance, reconstructions of historical

floods of the last 200 years in the Ahr valley identified events in 1910 and 1804 with numerous fatalities and the flood peak of 1804 was very similar to that of 2021 (Vorogushyn et al., 2022; Fig. 5). However, when past events are too far back in time, they are usually not remembered.

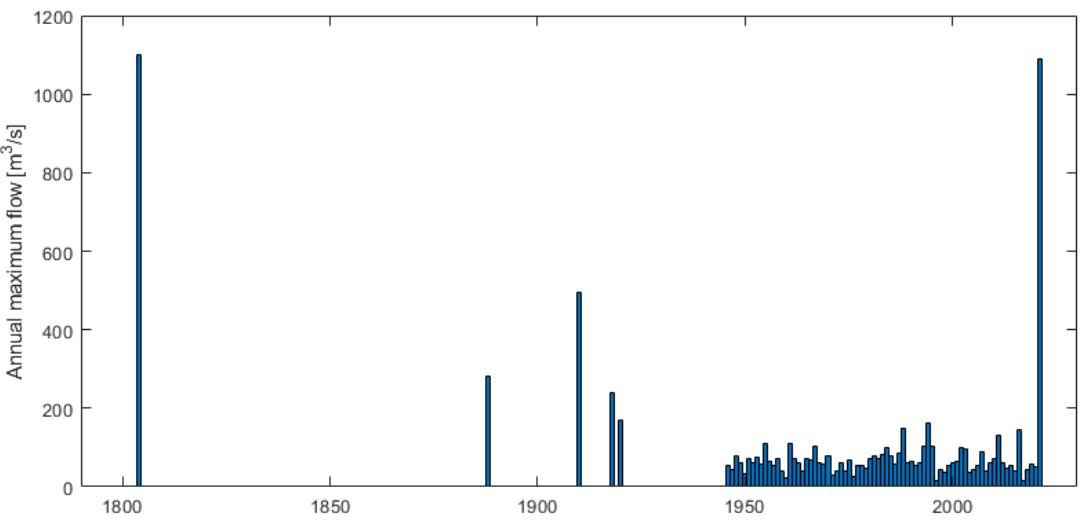

**Figure 5: Time series of the annual maximum flow of the Ahr River at Altenahr from 1804 to 2021. The five historic flood peaks have been reconstructed by Roggenkamp and Herget (2014). Systematic data: Landesamt für Umwelt Rheinland-Pfalz**

### 3.3 Socio-economic reasons

There are numerous socio-economic reasons for the belief that an event is impossible, or for not being fully prepared, including psychological, political and economic reasons. In fact, "Individuals show boundedly rational behaviour" (Aerts et al., 2018). They tend to overestimate their risk directly after a flood, but underestimate risk in periods without flood experience. As a consequence, they tend to postpone mitigation measures, e.g. constructing flood-proof windows or storing keepsakes and important documents on upper floors (see the overview of Kuhlicke, 2020). The decision of whether or not to prepare communities for an event, and thus to not consider it impossible, is also part of a political strategy, since it involves costs to the community. It is a decision that is often unpopular because it does not translate into immediate benefits to citizens but into immediate costs. It is also a matter of preparing for a rare event that is unlikely to occur in the short span of an administrative term. The management of such decisions, from a political point of view, also entails a cost in terms of visibility and personal popularity, and is therefore unprofitable for the political entity that has to promote and approve it. This is the reason why these decisions are often postponed until the next term of administrators. Exceptions are the circumstances immediately following a tragedy, as numerous recent flood defense actions demonstrate. In fact, governments tend to show reactive behaviour and respond as soon as a disastrous event has occurred instead of behaving proactively (Haer et al., 2019). Even when the political will arises, it is often problematic to intervene due to a lack of resources. This problem is obviously more common in countries with large public debt. (Knittel et al., 2023).

## 3.4 Combined and concurrent reasons

Often it is not only physical, psychological or socio-economic reasons that lead to underestimating the risk, but rather a combination of them. The driving forces of extreme floods may interact with each other with poorly predictable dynamics, especially when it comes to feedbacks between the hydrological and social systems (Sivapalan et al., 2012). For example, after levees are raised to protect flood-prone areas, people tend to feel safer inducing socio-economic development and therefore increased exposure which, in turn, will lead to a further raising of the levees if an event occurs that exceeds the levee height. This happened at many large rivers around the world such as the Po and the Danube (Di Baldassarre et al., 2009; D'Angelo et al., 2020). Another example of combination of reasons is given by fire-flood interactions. In 2019/2020, wildfires ravaged ecosystems and settlements in Australia after a severe drought, which reduced surface infiltration capacity and intensified flooding and erosion, resulting, among other things, in siltation of Sydney's main water supply reservoirs (Kemter et al., 2021, Xu et al., 2023). A recent real-world example is the flood that occurred in Lybia in September 2023. It was caused by the combination of a severe rainstorm on land with sparse vegetation cover and low retention capacity, and the collapse of two dams that unleashed a catastrophic flood wave. About a quarter of the city of Derna was destroyed, causing the death of thousands of persons.

If two concomitant factors A and B are independent, the probability of both occurring, $P(A \cap B)$, is given by the product of the two marginal probabilities $P(A)$ and $P(B)$. If, on the other hand, they are not independent, $P(A \cap B)$ is given by the product of $P(A)$ and the probability of B conditioned by the occurrence of A, $P(B|A)$. The latter may be much higher than $P(B)$, and therefore neglecting dependence may lead to underestimating $P(A \cap B)$. For instance, about 190,000 people living downstream of the Oroville dam in California had to be evacuated in February 2017 when extreme streamflow damaged its spillways leading to a rapidly developing emergency situation (Vahedifard et al., 2017) (factor A). When the emergency spillway was used instead, headward erosion threatened to undermine and collapse the concrete weir (factor B). While the original design counted on the redundancy of the system related to statistically independent failure of spillways probabilities, the causes of the two failures were actually similar, high flows and lack of maintenance in both cases, resulting in a much higher chance of failure of both spillways, in a way that a 2-year return period rainfall event combined with high antecedent soil moisture could cause this hazardous situation.

## 4 The importance of assessing the impact of the impossible flood

The feeling that an extreme flood is impossible may therefore imply ignoring part of the flood risk. The decision to ignore a risk component is not necessarily unreasonable: since full security against a flood event is never guaranteed, the community will necessarily have to accept taking a risk, but should be aware of it. We have previously emphasised how the impossible flood is often overlooked without being aware of the associated risk, thus exposing the community to an unconscious dangerous situation, i.e. mirroring the situation of Damocles' banquet. In order to understand the peculiarity of this situation, it is necessary to recall that risk is defined as potential loss of life, injury, or destroyed or damaged assets which could occur to a system, society or a community in a specific period of time, determined probabilistically as a function of hazard,

exposure, and vulnerability (UNISDR, 2017; Klijn et al., 2015). Catastrophic and unexpected events may involve multiple and diverse losses of tangible and intangible values. In particular, associating a monetary price to human life and other social values like cultural heritage may be not advisable when assessing the risk associated to large disasters, as there might be significant moral and ethical implications (Kellert, 1996; Kuchyňa, 2015). Moreover, social vulnerability is recognised to play a relevant and specific role in flood risk management. It may assume a diverse set of spatially and temporally variable connotations (Koks et al., 2017) thus suggesting that its assessment should be separate from that of physical vulnerability. Accordingly, we propose here to assess economic risk and social risk separately.

The time averaged economic risk, ED, in monetary units, arising from a natural event is typically represented by the Expected Damage *ED* estimated as (Klijn et al., 2015)

$$ED = \int_{x_d}^{\infty} p(x)V_e(x)E_e(x)dx \tag{1}$$

where $x_d$ is the intensity of the event above which damage occurs (such as the safety level of flood defences), $p(x)$ is the probability density of the event with intensity $x$, $V_e(x)$ is economic vulnerability, and $E_e(x)$ is exposure. Here, we define vulnerability as the degree of protection of the assets exposed to the risk which varies from 0 to 1. $V_e(x) = 0$ implies maximum protection, while $V_e(x)=1$ implies no protection. Protection is understood in a comprehensive way, including structural and non-structural measures that affect the damage of an event, such as forecasting and early warning systems or communities with high social capital where people support each other in emergency situations. We define exposure $E_e(x)$ as the economic damage caused by the event [monetary units].

The expected social risk, ES, can be estimated as

$$ES = \int_{x_d}^{\infty} p(x)V_s(x)E_s(x)dx \tag{2}$$

where $x$ and $p(x)$ have the same meaning as in Eq. (1), and $V_s(x)$ and $E_s(x)$ are social vulnerability and social exposure, respectively, measured with appropriate units. For instance, Tate et al. (2021) estimated social exposure in terms of number of people exposed to flooding. In general, $V_s(x)$ and $E_s(x)$ can be estimated based on social features and behaviors (Koks et al., 2017; Sorg et al., 2018; Spielman et al., 2020).

The impossible flood is characterised by low $p(x)$ values and thus its contribution to risk may be very small. Systematic analyses based on typical values of flood frequency behaviour, flood plain morphology etc. suggest that events with a return period of more than 1,000 years contribute only around 1% to the risk (Merz et al., 2009). This small percentage is in line with the observation that high impact events are usually not considered very relevant, which is in stark contrast with its societal relevance and human perception in the case such an event occurs. One incident with 1,000 fatalities is normally perceived as much more important than 1,000 incidents with 1 fatality each (Kasperson et al., 1988), indicating that the statistical evaluation does not mimic human perception well. Society tends to dread high impact/low probability events, and

240 there are good reasons for this risk aversion. The impossible flood can substantially affect the long-term development of regions and communities, and is often associated with follow-up effects, e.g. post-traumatic stress disorder, that extend far beyond the direct effects that are usually accounted for in risk analyses. Thus, economic indirect impacts might be comparable to direct damages (as for the 2011 floods in Thailand discussed by Merz et al., 2015).

Since the impossible flood is a very intense event, vulnerability tends to unity while exposure could assume a surprisingly
high value, especially if the flood is caused by a combination of factors. In other words, should the event occur, the economic and especially the social damage, including loss of life, could be enormous. Therefore, rather than relying on its very low probability through the risk equation, it may be appropriate to take into account risk aversion, which implies that communities may be willing to adopt policies to mitigate the damage caused by very extreme events irrespective of their low probability (Kind et al., 2016). In such cases, it might be advisable to refer to the probable maximum loss, which is widely
used in the insurance sector, defined as the largest loss that an insurance company might face. During the banquet, Damocles certainly did not estimate the probability of the horsehair to break. Instead, he more emotionally admitted that the inauspicious event could occur. He was concerned about his own safety and left the room to avoid taking the risk of losing his life, however small. In order to avoid situations such as that of Damocles in the context of flood hazard, a higher priority needs to be attributed to very extreme floods when designing flood management strategies. This would imply using the risk
equation as usual, for example designing flood defence on the basis of cost-benefit analysis, but giving a greater weight to dreadful floods. Alternatively, or additionally, one needs to understand how impossible floods may potentially come about and their consequences with the aim of devising counter measures, while estimating their precise probability becomes less important.

## 5 Estimating magnitude and impact of the impossible flood

Because of its unusual nature, estimation of the impossible flood and its impact requires a broader perspective than that suitable for smaller floods. It may involve new methods for prioritising alternative scenarios that go beyond flood events with recurrence intervals of 100 or 200 years, which is the design return period usually recommended by flood directives. There are two approaches of addressing the prioritisation of risk management measures (Fig. 6). The first, traditional one, termed the top-down approach, starts from the climate forcing, cascading down information to the people affected by the
floods. The second, termed bottom-up approach, starts from the local scale of individuals, households and communities and explores the factors and conditions that enable successful coping with floods (Wilby and Dessai, 2010; Blöschl et al., 2013). The bottom-up approach is motivated by a social paradigm. Hence risk is not only defined in monetary terms but also includes a social component. The main goal is not to find the most economic management strategy but to ensure the wellbeing of people by reducing vulnerability and enhancing resilience (the ability to recover after an event). It does not take
climate projections as a starting point but the vulnerability and resilience of the risk related system itself (van Pelt and Swart, 2011). It is sometimes termed the "assess-risk-of-policy" method as it explores alternative policies first. While the top-down approach is currently more popular, realisation of the importance of the bottom-up approach is emerging: "Society will even

benefit much more from a greater understanding of the vulnerability of climate-influenced decisions to large irreducible uncertainties than it will from extremely expensive attempts to increase the accuracy and precision of climate predictions. An alternative approach to the conventional one based on climate prediction would therefore focus on exploring how well strategies perform across wide ranges of assumptions and uncertainties (Robust Adaptation Decision-Making)." (European Commission, 2009, p. 13). Typically, the strategies are not optimal from an economical perspective but they are robust, i.e. they are designed to perform well over a wide range of assumptions including impossible floods.

The broader, bottom-up approach may include worst-case scenarios of failure of structural and organizational protection measures: What might happen if dikes break or warning chains fail? In addition to estimating flood peaks and inundation areas, the entire range of relevant processes would be considered, such as transport of bedload and dead wood, clogging of culverts, undercutting of bridges and buildings, landslides or release of chemicals. Such processes are not usually included in risk assessments today because of data limitations and limited scope, while a lot of effort is invested in hydrological and hydrodynamic models to reproduce the observed flood behaviour as precisely as possible. However, this approach fails when the processes of the extreme situations differ from those commonly observed.

For example, during the 2021 Ahr flood, communication and electric power networks failed almost immediately, which hindered disaster management and aggravated the impact. In one of the hospitals the staff had to carry the patients through the staircase to evacuate them as the elevators did not work because the backup power supply of the hospital had also failed. The Thailand 2011 flood interrupted the global supply chains of the car industry with significant economic consequences. Prior to this flood, this increase in risk due to the economic development of the floodplains of the Chao Phraya River and its pivotal role for the global car industry had not been recognized.

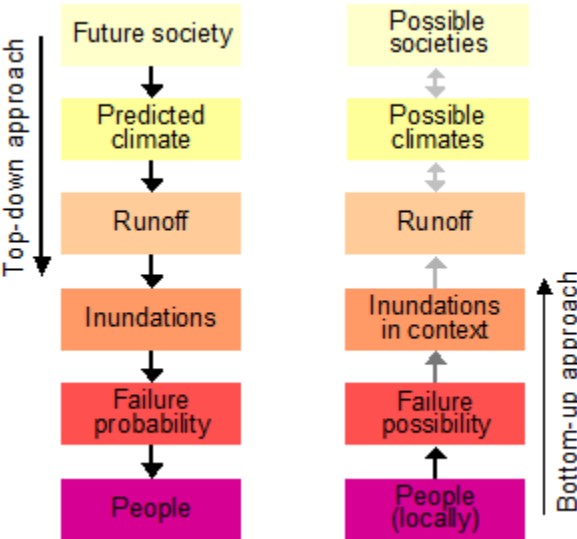

**Figure 6: Left: Traditional top-down approach to flood risk assessment based on climate projections. Right: Bottom-up approach to hydrological risk assessment that is vulnerability or resilience centred. Grey arrows indicate less dependence than black arrows. From Blöschl et al. (2013).**

While hydrologists have developed a wealth of approaches to estimate small to moderately extreme floods, i.e. from the 2-year to the 200-year flood, process studies on the impossible flood are relatively rare. We argue that this blind spot should be addressed more systematically than in the past. When exploring the impossible flood, even models that have large uncertainties could provide valuable guidance. Expert knowledge and semi-quantitative methods can be used to compensate for missing observational data and process models. This general approach of extending the information base in flood estimation, including for extreme floods, is known as the flood frequency hydrology framework (Viglione et al., 2013) and is recommended in the German and Austrian flood estimation standards (DWA, 2012; ÖWAV, 2019).

Within this general framework, there is a spectrum of possible approaches, ranging from fully process-based, probabilistic methods to semi-quantitative, simplified system representations. This spectrum includes:

- Extrapolation from frequent to impossible floods: careful attention needs to be paid to the processes at the upper tail of the distribution (Nguyen et al., 2024). While, traditionally, floods peaks are assumed independent, identically distributed random variables, this assumption may no longer hold if the impossible flood is dominated by processes other than the majority of smaller floods, so the change of processes needs to be explored.

- Stochastic simulation, by embedding process models in a stochastic environment and generate large sets of events from which impossible floods are derived: while in the past these techniques were often limited to Gaussian and independent processes, recent developments allow simulations of a wide range of dependence structures and underlying probability distributions (Montanari et al., 1997; Papalexiou et al., 2021).

- Perfect Storms, i.e., identifying the most unfavourable (but possible) superposition of processes (Paté-Cornell, 2012): an example is the superposition of the event precipitation of past floods with the initial catchment soil moisture of other observed floods with smaller rainfall but wetter initial conditions. Another possibility is to shift the precipitation field of observed floods in space to explore whether physically plausible shifts would lead to higher peaks, or to maximise process intensities based on physical considerations.

- Reforecast Simulations, i.e. ensemble forecasts generated for past periods, for generating more extreme floods that those observed (Kelder et al., 2020, Brunner and Slater, 2022): For instance, outliers in the ensemble of past precipitation forecasts could be used as forcings to flood models for simulating unprecedented flood events.

- Storylines, i.e. plausible flood scenarios based on expert knowledge: for instance, Albano et al. (2016) applied an extreme winter storm scenario for California (a concatenation of two historic atmospheric river storm sequences from 1969 and 1986), to explore the potential impacts from extreme storms to the emergency management life cycle. Such an event would overwhelm California's flood protection on a large scale (Porter et al., 2010).

- Downward counterfactuals, i.e. thinking about what did not happen but could have happened: past events (disasters or near-disasters) are modified to develop scenarios in which the event turns out to be worse than what had actually happened. This method may be able to identify Black Swans, i.e. events that are rare, take people by surprise and have huge impacts (Woo, 2019).

## 6 Measures to prepare against the impossible flood

When having estimated the impossible flood and the associated risk, the question arises how society can prepare for such an event. It must first be realised that flood management measures are usually different from those for minor floods. In particular, it is desirable to set up a diverse portfolio of different types of risk reduction measures operating at multiple spatial scale, that targets events of different return periods. Such approach is known as "risk layering", a concept used in many areas of risk policy (FONDEN, 2012). This approach allows identifying risk management options that are specifically effective for low, medium and high-return period events.

The reasons why dreadful floods are considered impossible discussed in Section 3 can provide guidance for the identification of the most appropriate solution to mitigating their impact. Some examples are given here below.

### 6.1 Physical/technological tools:

- Assessment of the social and economic damage caused by the impossible flood in the event of its occurrence. This implies careful analysis of the tail end of the exposure curve in the low probability range;

- Design of structural flood risk mitigation interventions with consideration of the hypothetical scenario induced by the impossible flood. The aim is to design redundant safety systems and flood management plans that consider floods with a return time much higher than that required by current regulations. Safety systems subject to a threshold effect are to be avoided (a well-known example is provided by morning glory spillways for reservoirs, whose performance suddenly drops when the outlet becomes clogged and outflow takes place under pressure);

- Fail-safe design of flood defence structures that avoids failure of the whole defence system if part of it fails (e.g. floodplain compartmentalization; Alkema and Middlekoop, 2005; Oost and Hoekstra, 2009). This scheme requires extensive discussion with stakeholders as it deliberately distributes the risk among different locations;

- Generation of future climate scenarios with particular attention to the tail end of the probability distribution, even though it is associated with great uncertainty;

- Interventions to reduce both social and economic exposure. This scheme also requires extensive discussion with stakeholders;

- Intensification of riverbed and levee maintenance operations, using methods that anticipate the occurrence of the impossible flood, to avoid crises such as that of Oroville in 2017 (Hollins et al., 2018);

- Improvements of flood forecasting and early warning: in case an impossible flood occurs, a main target is to prevent fatalities and other severe health consequences, such as injury, psychological trauma. Flood forecasting may be extended to smaller catchments and include impact forecasting (Apel et al., 2022).

## 6.2 Training and education tools:

- Training citizens exposed to floods and policy/decision makers with scenario-based approaches inspired by the storyline of familiar events, which are more likely to resonate with stakeholders themselves and are more likely to be considered plausible (Vasileiadou and Botzen, 2014; Alexander, 2000). Training may help to reduce recency bias by focusing on relevant past events, such as the 1804 and 1910 floods in the Ahr Valley (see Fig. 5);

- Flood drills, which too contribute to reducing cognitive biases and building the skills needed for defense and 365 evacuation measures to be carried out effectively in an emergency case;

- Reinforcement of higher education initiatives associated to specific training. Recent studies suggest that flood early warning is effective in reducing impact only when people know how to react when they receive a warning (Kreibich et al., 2021);

- Raising public awareness of the risk from events that are considered impossible, promoting greater awareness of 370 both the danger and the tools for assessment and mitigation.

## 6.3 Political and economic instruments

- Funding of both applied and basic research projects ('blue sky research'). Knowledge about extreme floods is still limited and there is a need to improve scientific understanding of the hydrological impacts of climate change and other drivers of change, e.g. socio-economic development, including the long-term interactions between humans 375 and floods (Sivapalan et al., 2012);

- Promotion of better synergy between academia and research institutions on the one hand, and political and spatial governance institutions on the other. Too often knowledge is not fully taken into account, partly for psychological reasons. The scientific community may play a key role by delivering a transparent and agreed upon message on the risks associated to impossible events;

- Incentives to support the reduction of exposure, e.g. through relocation;

- Funding of flood defences, and educational, warning and governance tools to cope with exceptional floods.

A second classification of solutions according to their effectiveness in mitigating different risks may provide useful support to the identification of their optimal combination in the resulting portfolio of measures. Figure 7 provides such classification by referring to the probability-damage diagram introduced in Fig. 3 to classify the different types of risk. Those measures 385 that respond to the Damocles risk type of the impossible flood are in the bottom right corner.

The effectiveness for specific local cases of flood risk reduction interventions should be verified by a quantitative assessment of their capability to reduce vulnerability and exposure against low probability events. Figure 8 illustrates the concept by showing the effect on vulnerability and exposure of two types of intervention as a function of flood return period. Structural interventions (e.g. the construction of levees and dams, Fig. 8 left) reduce the vulnerability over a limited range of return 390 periods, while being ineffective - or having even negative effects - for return periods higher than the design return period. When a levee is built, the ensuing development increases exposure behind the levee also for small floods. However, this is

not a problem because vulnerability is small, as the presence of the levee increases the degree of protection of the land behind the levee. It is only beyond the design flood of a levee that the increased exposure combines with increased vulnerability, leading to an overall large increase in risk.

**Figure 7: Effectiveness of flood protection measures in mitigating the risk associated to events with different probability of occurrence and extent of damage.**

Figure 8 right shows an example of an effective measure against an impossible flood, the reduction of exposed values

through urban planning. The drawback, however, is that it may be impossible to take measures as social (political and economic) costs may be excessively high. Thus, choices may require a long-term careful planning in order to avoid 'lock-in' situations, i.e. stalling of the socio-economic system in a sub-optimal state (Sivapalan and Blöschl, 2015), due to feedbacks, e.g., related to lack of resources or public resistance to adopt any solution. In this sense, the attitude of the scientific community and institutions is again crucial. In particular, the scientific community may enable synthesis efforts by issuing

agreed upon technical messages stressing the importance of a long-term planning perspective of sharing information transparently and striving for objectivity in reporting.

## 6 Conclusions

Floods associated with very low probabilities of occurrence are often considered impossible. As a consequence, their impact may not be fully perceived by citizens, experts and politicians, and their risk may be underestimated. Because of this,

dreadful floods often take communities by surprise. While the decision to prepare or not for such an event is a political one, awareness of its potential occurrence is essential, thus paying close attention to the impact of a flood that exceeds in magnitude (or other characteristics) those accounted for by current engineering codes. It is necessary to assess the vulnerability and exposure of the communities with respect to the impossible flood. An exposure curve that increases excessively at high return periods would indicate an undesirably high economic and social cost, even if its contribution to the

average risk is low.

This  paper reviews methods for assessing the risk of 'impossible floods', which should include bottom-up approaches that start from the people affected and explore risk management options, giving high priority to social in addition to economic risks. Suggestions for managing this risk are given, with a focus on measures specifically geared towards  impossible floods. Protecting oneself from these floods may prove too costly and place too great a burden on society. In such cases, education and awareness raising are among suitable instruments to avoid a major disaster.

The scientific community plays a key role in achieving this by delivering a transparent and objective message highlighting the actual risk associated with impossible floods and potential solutions for its mitigation. Impossible floods are becoming increasingly possible and therefore increasingly threatening. This calls for a shift from crisis response to proactive, long-term flood risk management, a renewed focus on creative exploration of hazard scenarios and risk management strategies, and on communicating the unknown corners of the reality of floods.

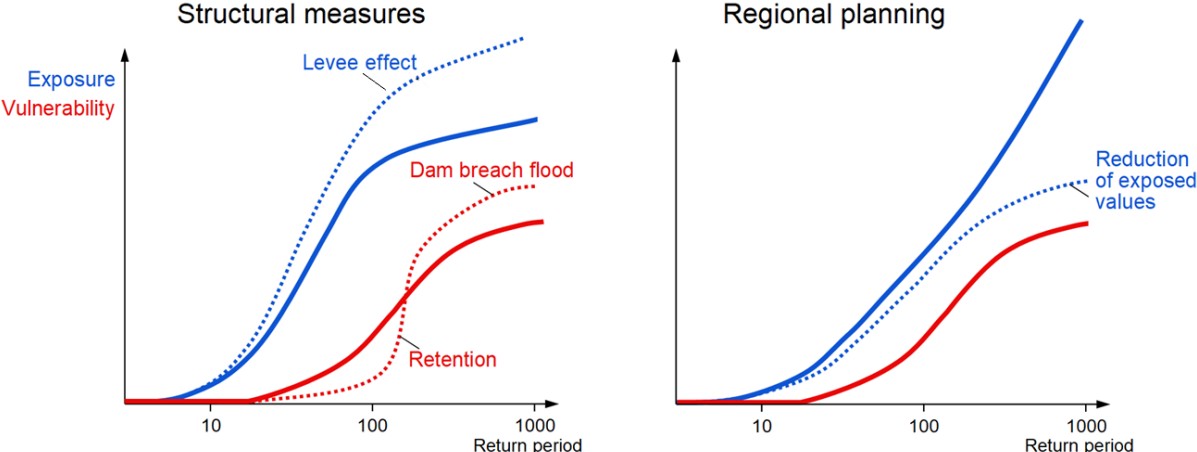

**Figure 8: Example of exposure and vulnerability as a function of flood return period downstream of structural interventions (left), and urban planning to reduce exposure (right). Exposure reduction is a robust strategy for reducing the risk from an extended range of flood discharge. Events considered impossible are on the right-hand side of the figures.**

## Competing interests

At least one of the coauthors is a member of the editorial board of Hydrology and Earth System Sciences.

## Acknowledgements

Funding by the Austrian Science Funds (project Spate, I 3174, I 4776), DFG Research Group FOR 2416 'Space-Time Dynamics of Extreme Floods – SPATE', grant number 278017089), Federal Ministry of Education and Research of Germany (BMBF KAHR project; grant number 01LR2102A), and European Union Next-GenerationEU (National Recovery and Resilience Plan – NRRP, Mission 4, Component 2, Investment 1.3 – D.D. 1243 2/8/2022, PE0000005) is acknowledged.

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
