# Peer review of "HESS Opinions: The Sword of Damocles of the Impossible Flood"

_EGUsphere, 2023_

## Author Comment (AC1)

**Reply to referee #1**

We thank the Referee for the positive premise of the review. Indeed, it is our aim to present a manuscript that is a "joy to read", as capturing the interest of the audience is an essential ingredient for disseminating awareness of the risk associated to impossible floods. We appreciate the insightful comments of the Referee, which we individually discuss below along with the proposed changes. Comments of the reviewer are copied in italic, while text that we suggest to amend/add in the manuscript is marked in red.

*Section 2, lines 38-41: The authors state that "communities often design their management strategies based on historical events" and such "impossible events include floods whose probability of occurrence is considered too small to act, but also events that go beyond any imagination." Additionally, large flood events may be not covered by underlying regulations – the design events authors refer to later on are an example, as e.g. given in the EU Flood Directive (Commission of the European Communities, 2007) and related national law. Authors are kindly requested to make a few statements here.*

We agree that regulations often do not require to assess the risk associated to floods with return period larger than 100-200 years. We recognize in the manuscript that this is one of the reasons why impossible floods are often not perceived as a significant hazard. We agree that a reference to the EU Flood Directive (Commission of the European Communities, 2007) should be included in Section 5. Moreover, we will add the following sentence in the first paragraph of Section 2, between the first and second sentence (line 40):

"In fact, international and national regulations recommend design recurrence intervals that usually do not exceed 100 or 200 years, while impossible floods are far less frequent. For example, the EU Flood Directive (Commission of the European Communities, 2007) requires member states to map inundation areas for an extreme scenario which has typically a return period of 200 years."

*Section 2, line 50f.: The challenge of dynamics in discharge (observation in the past ≠ system behaviour in the future) have been extensively discussed (see e.g., Sofia et al., 2017). The authors may wish to comment on their assumption that this is not considered in contemporary flood risk management, also against previous works of the authors of the present manuscript (Blöschl et al., 2019; Blöschl, 2022).*

We agree that changing dynamics of river discharge may induce underestimation of the risk associated to extreme events. In fact, we already state in line 52 that "However, in reality the future may be quite different from the past, not least because of climate change and our limited knowledge of the past itself." We will include citations to the relevant literature including Sofia et al. (2017), Blöschl et al. (2019) and Blöschl et al. (2022).

*Section 3, line 140: What is meant here with the term "residual risk"? Failure of technical mitigation? Events larger than expected? The issue of not remembering historical (or even younger) flood events and associated losses has been a topic for decades (Kuhlicke et al., 2020), however, experienced flood damage in the past does not necessarily mean good governance in the future. The authors may wish to comment on these issues specifically in terms of the "recency bias" (line 141).*

We agree that a definition of what we mean by "residual risk" is needed here and change the text of line 136-137 as follows:

"Single-action bias: Relying on one action, even when this action reduces the risk only marginally. Therefore, a significant risk (residual risk) remains that is often not perceived (CRED, 2009). Residual risk is defined as the risk that remains, even when effective disaster risk reduction measures are in place, and for which emergency response and recovery capacities must be maintained (UNISDR, 2009; Fiori et al., 2023)".

Regarding the recency bias, we will modify the sentence at line 130 as follows:

"Recency bias: Overemphasizing the importance of recent events, e.g. associating a higher probability of occurrence with more recent events, even when information of large past events is available".

*Section 3, line 151f.: The decision of whether or not to prepare for an event is not necessarily (or only) a political one. Fellows from social sciences have a long tradition in studying the willingness of homeowners to prepare for undesirable events by e.g. local structural protection or retrofitting of elements at risk. Examples from Europe include but are not limited to the overview of Kuhlicke (2020) or Attems et al. (2020). As such, the authors may wish to expand their vision here.*

In our manuscript we emphasized at 159 – 161 that "Similar behavioural patterns are found at the level of individual households and businesses. Individuals show boundedly rational behaviour (Aerts et al., 2018). They overestimate their risk directly after a flood, but underestimate risk in periods without flood experience and tend to postpone mitigation measures." However, we recognize that the train of thoughts in Section 3.3 needs to be revised to make it more coherent. We propose to modify the Section as follows (lines 150-164):

"There are numerous socio-economic reasons for the belief that an event is impossible, or for not being fully prepared, including psychological, political and economic reasons. In fact, "Individuals show boundedly rational behaviour" (Aerts et al., 2018). They tend to overestimate their risk directly after a flood, but underestimate risk in periods without flood experience and tend to postpone mitigation measures, e.g. constructing flood-proof windows or storing keepsakes and important documents on upper floors (see the overview of Kuhlicke, 2020). The decision of whether or not to prepare communities for an event, and thus to not consider it impossible, is also part of a political strategy, since it involves costs to the community. It is a decision that is often unpopular because it does not translate into immediate benefits to citizens but into immediate costs. It is also a matter of preparing for a rare event that is unlikely to occur in the short span of an administrative term. The management of such decisions, from a political point of view, also entails a cost in terms of visibility and personal popularity, and is therefore unprofitable for the political entity that has to promote and approve it. This is the reason why these decisions are often postponed until the next term of administrators. Exceptions are the circumstances immediately following a tragedy, as numerous recent flood defense actions demonstrate. In fact, governments tend to show reactive behaviour and respond as soon as a disastrous event has occurred instead of behaving proactively (Haer et al., 2019). Even when the political will to intervene arises, it is often problematic to intervene due to a lack of resources. This problem is obviously more common in countries with large public debt."

*Section 4, the concept of risk: Surprisingly the authors make a distinction here between "economic risk" (eqn. 1) and "social risk (eqn. 2), which is not made clear in the text. Consistent with ISO Guide 73 and ISO 14091 (International Standards Organisation, 2009, 2021) and the UN/IDRR terminology on disaster risk reduction (UNISDR, 2017), risk is defined as "potential loss of life, injury, or destroyed or damaged assets which could occur to a system, society or a community in a specific period of time, determined probabilistically as a function of hazard, exposure, vulnerability and capacity". As such, exposure (line 201) cannot be defined as "the economic damage caused by an event", it should be the overall value of exposed elements at risk, reduced by the damage ratio (as a factor coming from the vulnerability part of the risk equation). Having said this it remains unclear what is meant by "social exposure" (line 206), and how social vulnerability can be conceptualised here (see e.g., Sorg et al., 2018; Spielman et al., 2020).*

Indeed, we suggest that social risk is assessed by using non-economic measures, to avoid that an economic value is associated to intangible values (see Koks et al. (2017) which we would like to cite in a revised version of the manuscript). In particular, we believe that associating an economic value to human life is inappropriate for extreme catastrophes that may cause a large number of life

losses. We agree that our motivation to assess economic and social risk separately has not been discussed in the text. Therefore, we propose to modify the wording in section 4 (line 189 and following ones) as follows:

"In order to understand the peculiarity of this situation, it is necessary to recall that risk is defined as potential loss of life, injury, or destroyed or damaged assets which could occur to a system, society or a community in a specific period of time, determined probabilistically as a function of hazard, exposure, and vulnerability (UNISDR, 2017; Klijn et al., 2015). Catastrophic and unexpected events may involve multiple and diverse losses of tangible and intangible values. In particular, associating a monetary price to human life and other social values like cultural heritage may be not advisable when assessing the risk associated to large disasters, as there might be significant moral and ethical implications (Kellert, 1996; Kuchyňa, 2015). Moreover, social vulnerability is recognised to play a relevant and specific role in flood risk management. It may assume a diverse set of spatially and temporally variable connotations (Koks et al., 2015) thus suggesting that its assessment is separated from that of physical vulnerability. Accordingly, we propose here to assess economic risk and social risk separately."

Furthermore, we propose to clarify in the text how social vulnerability can be conceptualized by making reference to the work of Koks et al (2015). Accordingly, we propose to modify the text at lines 200 and 201 as follows:

"The expected social risk, ES, can be estimated as

$$ES = \int_{x_d}^{\infty} p(x)V_s(x)E_s(x)dx$$

where $x$ and $p(x)$ have the same meaning as in Eq. (1), and $V_s(x)$ and $E_s(x)$ are social vulnerability and social exposure, respectively, measured with appropriate units. For instance, Tate et al. (2021) estimated social exposure in terms of number of people exposed to flooding. In general, $V_s(x)$ and $E_s(x)$ can be estimated based on social features and behaviors (Koks et al., 2017; Sorg et al., 2018; Spielman et al., 2020).

*Section 4, lines 2019ff.: The authors argue that exposure and vulnerability will be extremely high in case of "impossible flood events" – which is not a contradiction of the risk concept. It depends on the way risk is expressed, in terms of annual risk as in many cost-benefit analyses (then risk is relatively small) or in terms of probable maximum loss (then the values are relatively high). The authors may wish to expand their vision here.*

We thank the reviewer for the suggestion. We may reword the text at lines 220-222 as follows:

"In other words, should the event occur, the economic and especially the social damage, including loss of life, could be enormous. Therefore, rather than relying on its very low probability through the risk equation, it may be appropriate to take into account risk aversion, which implies that communities may be willing to adopt policies to mitigate the damage caused by very extreme events irrespective of their low probability (Kind et al., 2016). In such cases, it might be advisable to refer to the probable maximum loss, which is widely used in the insurance sector, defined as the largest loss that an insurance company might face"

*The same is valid for their argumentation in line 360ff. on levees, increasing exposure and vulnerability. It is not made clear why "vulnerability is small" if exposure increases behind a levee – social vulnerabilities may be the same or even higher, economic vulnerabilities depend on the susceptibility of elements at risk (as physical vulnerability does), and the institutional dimension of vulnerability may even be higher (see e.g. the discussion in Papathoma-Köhle et al. (2021)). The authors are kindly invited to clarify their statement.*

We define vulnerability as (see text in the manuscript at line 197) "as the degree of protection of the assets exposed to the risk which varies from 0 to 1". According to the above definition, a levee

implies a reduced vulnerability (provided the levee is not overtopped), regardless of the exposure behind the levee. To better clarify the concept we may modify the text at lines 361 and 362 as follows:

"When a levee is built, the ensuing development increases exposure behind the levee also for small floods. However, this is not a problem because vulnerability is small, as the presence of the levee increases the degree of protection of the land behind the levee. It is only beyond the design flood of a levee that the increased exposure combines with increased vulnerability, leading to an overall large increase in risk."

Once again, we would like to thank the reviewer for the very constructive assessment and suggestions.

**References**

[revised manuscript text omitted]

---

## Author Comment (AC2)

**Reply to referee #2**

We thank the Referee for finding the paper engaging and well written, and appreciating the metaphor of the Damocles' Sword, which we hope may deliver an effective message.

We provide here below our replies to the comments of the Referee, which are copied in italic.

*In the abstract and in other sections, the Authors seem to use as synonyms "impossible floods" and "mega-floods", but I think this can be confusing for the reader. Some countries like India and Bangladesh experience almost every year widespread flooding, and the size of related impacts would easily qualify them as mega-floods in many other countries, yet they far from being considered impossible. Conversely, more localized floods might have not so large impacts but still be regarded as "impossible".*

We agree with the reviewer. We will substitute the term "mega-flood" with the term "impossible flood" or "dreadful flood". We believe these terms better convey the intended meaning.*Section 3: the discussion could be more compelling by presenting more examples of past floods illustrating the reasons and the points made by the Authors. Ideally, examples should come from different continents (e.g. the 2022 floods in Pakistan might easily qualify as unexpected due to the sheer size and duration of the event; during the 2023 floods in Emilia-Romagna, Italy, multiple failures of flood defences caught by surprise the population in lowland areas; and the September 2023 floods in Lybia 2023 were a terribly fitting example of the Damocles metaphor).*

We thank the reviewer for the suggestion. We are willing to add a reference to the flood in Romagna at line 93, which reads:

"Another example is given by the flood that occurred in the Romagna region, in Italy, in May 2023. Here, a first extreme storm induced high antecedent soil moisture for a subsequent storm that caused extreme runoff and then extensive landslides and flooding."

Furthermore, we are willing to mention the 2023 flood in Lybia at line 174 by adding the text:

"A recent real-world example is the flood that occurred in Lybia in September 2023. It was caused by the combination of a severe rainstorm on land with sparse vegetation cover and low retention capacity, and the collapse of two dams that unleashed a catastrophic flood wave. About a quarter of the city of Derna was destroyed, causing the death of thousands of persons."

*Section 3.1. Another important reason that should be discussed here is socio-economic development (urbanization, floodplain development etc) which largely contributes to change exposure and vulnerability. Hence, past flood events that were moderately harmful might cause much larger damages, should they occur again under present conditions (see for instance the analysis by Paprotny et al, 2018)*

We fully agree with the reviewer. In fact, in section 3.4 lines 167-170 we provide the example of the levee effect, that precisely implies the increase in exposure that the reviewer mentions. To address the concern of the reviewer we propose to modify the relevant sentence in the paper as follows:

"For example, after levees are raised to protect flood-prone areas, people tend to feel safer inducing socio-economic development and therefore increased exposure which, in turn, will lead to a further raising of the levees if an event occurs that exceeds the levee height. This happened at many large rivers around the world such as the Po and the Danube (Di Baldassarre et al., 2009; D'Angelo et al., 2020)."

*Section 3.3. The reasoning here makes sense and I mostly agree, could you provide some references for these statements?*

We fully agree with above suggestion. We propose to include references to Kuhlicke (2020), Hausknost (2014) and Knittel et al. (2023) and change the text at line 151 – 164 as follows:

"There are numerous socio-economic reasons for the belief that an event is impossible, or for not being fully prepared, including psychological, political and economic reasons. In fact, "Individuals show boundedly rational behaviour (Aerts et al., 2018). They tend to overestimate their risk directly after a flood, but underestimate risk in periods without flood experience and tend to postpone mitigation measures, e.g. constructing flood-proof windows or storing keepsakes and important documents on upper floors (see the overview of Kuhlicke, 2020). The decision of whether or not to prepare communities for an event, and thus to not consider it impossible, is also part of a political strategy, since it involves costs to the community. It is a decision that is often unpopular because it does not translate into immediate benefits to citizens but into immediate costs. It is also a matter of preparing for a rare event that is unlikely to occur in the short span of an administrative term. The management of such decisions, from a political point of view, also entails a cost in terms of visibility and personal popularity, and is therefore unprofitable for the political entity that has to promote and approve it. This is the reason why these decisions are often postponed until the next term of administrators. Exceptions are the circumstances immediately following a tragedy, as numerous recent flood defense actions demonstrate. In fact, governments tend to show reactive behaviour and respond as soon as a disastrous event has occurred instead of behaving proactively (Haer et al., 2019). Even when the political will to intervene arises, it is often problematic to intervene due to a lack of resources. This problem is obviously more common in countries with large public debt. (Knittel et al., 2023)."

*L209-218 and 224-229: the authors might want to enrich their discussion by including the concepts of risk aversion and social vulnerability presented in previous works (e.g. Koks et al, 2015; Mechler, 2016; Kind et al 2017). By the way, it is worth mentioning here that low-probability, high-impact events are crucial also for the insurance sector, and this is reflected by the use of probable maximum loss (PML) as a standard metric for risk characterization.*

Reviewer #1 raised a similar remark on the way social vulnerability should be conceptualized and estimated. We would like to make reference to the work by Koks et al. (2015) that is cited by the reviewer, to clarify that social dimensions of risk – and spatial variation in those dimensions – indeed play a relevant and specific role in flood risk management, therefore providing ground for assessing social risk separately. Accordingly, we propose to modify the wording in line 189 and following ones by adding the statement:

"In order to understand the peculiarity of this situation, it is necessary to recall that risk is defined as "potential loss of life, injury, or destroyed or damaged assets which could occur to a system, society or a community in a specific period of time, determined probabilistically as a function of hazard, exposure, and vulnerability (UNISDR, 2017; Klijn et al., 2015). Catastrophic and unexpected events may involve multiple and diverse losses of tangible and intangible values. In particular, associating a monetary price to human life and other social values like cultural heritage may be not advisable when assessing the risk associated to large disasters, as there might be significant moral and ethical implications (Kellert, 1996; Kuchyňa, 2015). Moreover, social vulnerability is recognised to play a relevant and specific role in flood risk management. It may assume a diverse set of spatially and temporally variable connotations (Koks et al., 2015) thus suggesting that its assessment should be separate from that of physical vulnerability. Accordingly, we propose here to assess economic risk and social risk separately."

Furthermore we propose to modify the text at lines 200 and 201 as follows:

"The expected social risk, ES, can be estimated as

$$ES = \int_{x_d}^{\infty} p(x)V_s(x)E_s(x)dx$$

where $x$ and $p(x)$ have the same meaning as in Eq. (1), and $V_s(x)$ and $E_s(x)$ are social vulnerability and social exposure, respectively, measured with appropriate units. For instance, Tate et al. (2021) estimated social exposure in terms of number of people exposed to flooding. In general, $V_s(x)$ and $E_s(x)$ can be estimated based on social features and behaviors (Koks et al., 2017; Sorg et al., 2018; Spielman et al., 2020).

Regarding risk aversion and the use of the Probable Maximum Loss to assess flood impact, we propose to modify the text at line 220 – and following ones – as:

"In other words, should the event occur, the economic and especially the social damage, including loss of life, could be enormous. Therefore, rather than relying on its very low probability through the risk equation, it may be appropriate to take into account risk aversion, which implies that communities may be willing to adopt policies to mitigate the damage caused by very extreme events irrespective of their low probability (Kind et al., 2016). In such cases, it might be advisable to refer to the probable maximum loss, which is widely used in the insurance sector, defined as the largest loss that an insurance company might face."

*L216-218: Also, economic indirect impacts from severe floods (e.g. failures in transport and energy networks, disruption of business and production) might be comparable to direct damages (as in the example of 2011 floods in Thailand described by Merz et al., 2015)*

We propose to add the following sentence at line 218:

"Thus, economic indirect impacts might be comparable to direct damages (as for the 2011 floods in Thailand discussed by Merz et al., 2015)."

*L240-249: I beg the authors' pardon, but I'd like to play the role of the devil's advocate in the comparison between top-down vs bottom-up approaches. According to the discussion here, my impression is that the two approaches are alternative and the latter is preferable than the former, but is it always the case? Flood risk management is not exclusively a local problem and some sort of integration at (at least) river basin scale is needed, which includes also finding a compromise between contrasting local priorities, which has to be done as some intermediate level between top and bottom. For instance, raising dikes upstream can increase downstream risk, whereas flood detention areas comes at a cost for local communities in terms of restrictions even if this protects downstream communities. For these reasons, people and local stakeholders might be in favour of traditional flood control structures (e.g. dikes, river cleaning) and not see favourably alternative measures (river restoration etc) which are perceived as less effective and/or more costly. The authors might want to enrich the discussion by replying to these considerations.*

We indeed believe that in the case of flood management – and in particular the special case of impossible floods – the bottom-up approach is preferable with respect to top-down. We specified at line 234 and following ones that "There are two approaches of addressing the prioritisation of risk management measures ….. The first, traditional one, termed the top-down approach, starts from the climate forcing, cascading down information to the people affected by the floods. The second, termed bottom-up approach, starts from the local scale of individuals, households and communities and explores the factors and conditions that enable successful coping with floods….". Thus, with top-down approach we do not refer to the spatial scale, but rather to the cascade from climate models to hydrological models and consequential policies. With this approach, we believe that there is an amplification of uncertainties along the cascade that makes the final outcome in terms of copying with floods, less effective with respect to an approach that starts from the assessment of people's risk. The first step for the benefit of people is to get a more accurate assessment of flood risk, at the present time and the future. For the assessment of the most appropriate solution (that we discuss in the subsequent Section 6) we agree that a multi-spatial-scale assessment is needed.

To make the concept clearer, we propose to modify text at line 304 as follows:

"In particular, it is desirable to set up a diverse portfolio of different types of risk reduction measures operating at multiple spatial scales that targets events of different return periods."

*Section 4: how would you calculate in practice social risk? is there any past study that did something similar?*

There are several examples of studies where flood risk is assessed in non-economic terms (see for instance Tate et al., 2021) and social vulnerability is discussed (see Koks et al, 2015). To give an example, we propose to add a citation to Tate et al. (2021) at line 208 as follows:

"For instance, Tate et al. (2021) estimated social exposure in terms of number of people exposed to flooding."

*Section 5: Ideally, all listed actions should be accompanied by references to real-world applications, that would make the discussion more useful for the reader*

We included an extended set of references for each of actions we discuss. Some of the references refer to real world case studies, others refer to purely theoretical contributions. We added the reference to Nguyen et al. (2024) in the description of the first listed action.

Once again, we would like to thank the reviewer for the very constructive assessment and suggestions.

**References**

[revised manuscript text omitted]